Fine-mapping of qTGW2, a quantitative trait locus for grain weight in rice (Oryza sativa L.)

Zhang Hui 1 2 3
Zhu Yu-Jun 2
Zhu An-Dong 2
Fan Ye-Yang 2
Huang Ting-Xu 3
Zhang Jian-Fu 3
Xie Hua-An huaanxie@163.com 1 3
Zhuang Jie-Yun zhuangjieyun@caas.cn 2
1 College of Agriculture, Fujian Agriculture and Forestry University , Fuzhou , China
2 State Key Laboratory of Rice Biology and Chinese National Center for Rice Improvement, China National Rice Research Institute , Hangzhou , China
3 Rice Research Institute and Fuzhou Branch of the National Center for Rice Improvement, Fujian Academy of Agricultural Sciences , Fuzhou , China
Lu Kun
Electronic publication date: 2020 Mar 4
Publication date: 2020
Volume: 8
Electronic Location ID: e8679
Received 2019 Nov 15; Accepted 2020 Feb 3
Copyright: ©2020 Zhang et al.
Copyright year: 2020
Copyright holder: Zhang et al.
License: This is an open access article distributed under the terms of the Creative Commons Attribution License, which permits unrestricted use, distribution, reproduction and adaptation in any medium and for any purpose provided that it is properly attributed. For attribution, the original author(s), title, publication source (PeerJ) and either DOI or URL of the article must be cited.
License URL: https://creativecommons.org/licenses/by/4.0/

Keywords: Fine-mapping, Quantitative trait loci, Grain length, Grain weight, Grain width, Minor effect, Rice

Funding: Science and Technology Innovation Program of the Fujian Academy of Agricultural Sciences STIT 2017-1-1 National Research and Development Program 2016YFD0101801 Special Foundation of Non-Profit Research Institutes of Fujian Province 2018R1101013-4 National Natural Science Foundation of China 31521064 This work was supported by the Science and Technology Innovation Program of the Fujian Academy of Agricultural Sciences (STIT 2017-1-1), the National Research and Development Program (2016YFD0101801), the Special Foundation of Non-Profit Research Institutes of Fujian Province (2018R1101013-4) and the National Natural Science Foundation of China (31521064). The funders had no role in study design, data collection and analysis, decision to publish, or preparation of the manuscript.

==============================
Background

Grain weight is a grain yield component, which is an integrated index of grain length, width and thickness. They are controlled by a large number of quantitative trait loci (QTLs). Besides major QTLs, minor QTLs play an essential role. In our previous studies, QTL analysis for grain length and width was performed using a recombinant inbred line population derived from rice cross TQ/IRBB lines. Two major QTLs were detected, which were located in proximity to GS3 and GW5 that have been cloned. In the present study, QTLs for grain weight and shape were identified using rice populations that were homozygous at GS3 and GW5.

Method

Nine populations derived from the indica rice cross TQ/IRBB52 were used. An F10:11population named W1, consisting of 250 families and covering 16 segregating regions, was developed from one residual heterozygote (RH) in the F7generation of Teqing/IRBB52. Three near isogenic line (NIL)-F2 populations, ZH1, ZH2 and ZH3 that comprised 205, 239 and 234 plants, respectively, were derived from three RHs in F10:11. They segregated the target QTL region in an isogenic background. Two NIL populations, HY2 and HY3, were respectively produced from homozygous progeny of the ZH2 and ZH3 populations. Three other NIL-F2 populations, Z1, Z2 and Z3, were established using three RHs having smaller heterozygous segments. QTL analysis for 1000-grain weight (TGW), grain length (GL), grain width (GW), and length/width ratio (LWR) was conducted using QTL IciMapping and SAS procedure with GLM model.

Result

A total of 27 QTLs distributed on 12 chromosomes were identified. One QTL cluster, qTGW2/qGL2/qGW2 located in the terminal region of chromosome 2, were selected for further analysis. Two linked QTLs were separated in region Tw31911−RM266. qGL2 was located in Tw31911−Tw32437 and mainly controlled GL and GW. The effects were larger on GL than on GW and the allelic directions were opposite. qTGW2 was located in Tw35293−RM266 and affected TGW, GL and GW with the same allelic direction. Finally, qTGW2 was delimited within a 103-kb region flanked by Tw35293 and Tw35395.

Conclusion

qTGW2 with significant effects on TGW, GL and GW was validated and fine-mapped using NIL and NIL-F2 populations. These results provide a basis for map-based cloning of qTGW2 and utilization of qTGW2 in the breeding of high-yielding rice varieties.

Introduction

Rice is one of the staple food crops and consumed by half of the world’s population. Enhancing grain yield is always among main objectives of breeding programs. Grain weight is a key component of grain yield in rice, which is mainly determined by grain length, width and thickness. All these traits are quantitatively inherited and controlled by both major and minor genes.

More than 500 quantitative trait loci (QTLs) for grain weight and shape have been identified previously (http://www.gramene.org/). So far, 16 QTLs for these traits with large effects were cloned using diverse mapping populations. Among these, ten genes regulate grain weight by mainly affected grain length, including GS2/GL2, OsLG3, OsLG3b, GS3, GL3.1/qGL3, qTGW3/TGW3/GL3.3, TGW6, GW6a, GL6, and GLW7 (Fan et al., 2006; Qi et al., 2012; Zhang et al., 2012; Ishimaru et al., 2013; Hu et al., 2015; Song et al., 2015; Che et al., 2016; Si et al., 2016; Yu et al., 2017; Hu et al., 2018; Xia et al., 2018; Ying et al., 2018; Yu et al., 2018; Wang et al., 2019a; Wang et al., 2019b). Other four genes regulate grain weight by primarily influencing grain width, including GW2, GS5, GSE5 and GW8 (Song et al., 2007; Li et al., 2011; Wang et al., 2012; Duan et al., 2017). The remaining two genes, GL7/GW7, and GS9, have similar influences on grain length and width with opposite allelic directions, thus they hardly affect grain weight (Wang et al., 2015b; Wang et al., 2015c; Zhao et al., 2018). Identification of these genes have facilitated the breeding of high-yield rice varieties.

Although it has been recognized that both major and minor QTLs play essential roles in the genetic control of complex traits (Mackay, Stone & Ayroles, 2009), identification of minor QTLs has been limited due to their small effects across different genetic backgrounds and environments. Recently, more and more studies have paid attention to minor QTLs. A number of minor QTLs for grain weight and shape were fine-mapped, such as qTGW1.1a, qTGW1.2a, qGS1-35.2, qGW1-35.5, and qTGW10-20.8 (Zhang et al., 2016; Dong et al., 2018; Wang et al., 2019a; Wang et al., 2019b; Zhu et al., 2019b). Isolation of more minor QTLs would be beneficial for establishing the network regulating grain weight in rice.

In our previous studies, QTL analysis for grain length and width was performed using recombinant inbred lines (RILs) of indica rice crosses between Teqing (TQ) and IRBB lines. Two major QTLs were detected, which were located in approximate to GS3 and GW5, respectively (Wang et al., 2017). The present study aims to identify QTLs for grain weight and shape after eliminating the segregation of GS3 and GW5. Firstly, QTLs for grain weight and shape were detected using an F10:11 population that was derived from one residual heterozygote (RH) of TQ/IRBB52 and homozygous at GS3 and GW5. Then, a QTL region on chromosome 2 was selected for validation, dissection and fine-mapping. Using two sets of near isogenic lines (NIL) and six NIL-F2 populations derived from the F10:11 population, two QTLs were separated in the target region. One of them, qTGW2, was delimited into a 103-kb region flanked by Tw35293 and Tw35395.

Materials & Methods

Plant materials

Nine mapping populations of indica rice were used in this study. The first one was an F10:11 population that was previously developed from one RH in the F7 generation of TQ/IRBB52 (Zhang et al., 2019), consisting of 250 families and segregated 16 regions distributed on the 12 rice chromosomes. This population was named W1. The remaining eight populations were developed from RHs selected from the W1 population as described below and illustrated in Fig. 1. Three plants in F10, which carried heterozygous segments that covered partial or entire region of the interval Tw31911−RM266 on chromosome 2, were identified. They were selfed to produce three NIL-F2 populations consisting of 205, 239 and 234 plants and named ZH1, ZH2 and ZH3, respectively. Two QTLs were separated, of which qTGW2 located in the downstream region was selected for further analysis. Non-recombinant homozygotes were identified in the ZH2 and ZH3 populations and selfed. Two sets of NILs named HY2 and HY3 were developed, each consisting of 35 TQ homozygous lines and 35 IRBB52 homozygous lines. They were used to validate qTGW2. Then, other three RHs, carrying heterozygous segments overlapped in the terminal region Tw35293−RM266 of chromosome 2, were identified from the W1 population. The three plants were selfed to produce three NIL-F2 populations in F12, which consisted of 174, 237 and 228 plants and named Z1, Z2 and Z3, respectively.

Figure 1 Development of nine populations used in this study.

Field experiment and trait measurement

The rice populations were planted in the experimental stations of the China National Rice Research Institute located at either Lingshui in Hainan Province or Hangzhou in Zhejiang Province. For the F10:11 population and two sets of NILs, the experiments followed a randomized complete block design with two replications. For each replication, twelve plants per line were planted in one row. At maturity, five of the middle ten plants in each row were harvested in bulk and sun-dried. Two samples of approximately 10 g fully filled grains were randomly selected for the measurement of TGW, GL, GW, and length/width ratio (LWR) following the procedure reported by Zhang et al. (2016). For the six NIL-F2 populations, plants were harvested individually and sun-dried. Two samples of approximately 3 g fully filled grains of each plant were selected for the measurement of the four traits.

DNA marker analysis

For the W1 population and two NIL populations, leaf samples collected from the middle eight plants of a rice line were mixed for DNA extraction using a mini-preparation protocol (Zheng et al., 1995). For the six NIL- F2 populations, a two cm-long leaf sample collected from an F2 plant was used for DNA extraction using the same method. PCR amplification was performed according to Chen et al. (1997). The products were visualized on 6% non-denaturing polyacrylamide gels using silver staining. A total of 68 polymorphic markers were used, including 57 simple sequence repeats, eight insertion/deletions, one cleaved amplified polymorphic sequence, and two sequence tagged sites. Nine of them were developed according to sequence differences between TQ and IRBB52 detected with whole-genome resequencing (Table S1).

Data analysis

For the W1 and six NIL-F2 populations, genetic maps of each populations were constructed using Mapmaker/Exp 3.0, in which genetic distances between markers were presented in centiMorgan (cM) derived with Kosambi function. QTL mapping was performed using the default setting of the BIP (QTL mapping in bi-parental populations) approach in IciMapping V4.1 (Meng et al., 2015). LOD thresholds were calculated with 1,000 permutation test (P < 0.05) and used to claim a putative QTL.

For the two NIL populations, two-way analysis of variance (ANOVA) was performed to test the phenotypic differences between the two genotypic groups in each NIL set. The analysis was performed using the SAS procedure GLM (SAS Institute Inc, 1999) as described previously (Dai et al., 2008). Given the detection of a significant difference (P < 0.05), the same data were used to estimate the genetic effect of the QTL, including additive effect and the proportion of phenotypic variance explained (R2). QTL were designated according to the rules recommended by McCouch & CGSNL (2008).

Results

QTLs detected in the W1 population

A total of 27 QTLs for the four traits were detected, which were distributed on 14 segregating regions (Fig. 2, Table 1). Four of them had significant effects on three traits. In the Tw35293−RM266 region on chromosome 2, the IRBB52 allele increased TGW, GL and GW by 0.39 g, 0.060 mm and 0.013 mm, respectively. In the RM14032−RM14383 interval on chromosome 3, the IRBB52 allele increased GL by 0.025 mm, decreased GW by 0.014 mm and increased LWR 0.022. In the RM16252−RM335 region on chromosome 4, the IRBB52 allele decreased TGW and GW by 0.18 g and 0.017 mm, respectively, but increased LWR by 0.021. In the Tv963−RM27610 interval on chromosome 12, the IRBB52 allele increased GL by 0.031 mm, decreased GW by 0.013 mm and increased LWR 0.027. Worthy to note, the Tw35293−RM266 region had R2 of 14.50% for TGW and 18.60% for GL, which were much higher than the R2 values for these two traits detected in the other three regions.

Figure 2 Genomic distribution of QTLs for four traits detected in the W1 population.

TGW, 1,000-grain weight; GL, Grain length; GW, Grain width; LWR, Length/width ratio. The markers flanking GS2, GS3, and GW5 are indicated in boldface.

Table 1 QTLs detection for four traits using the W1 population.

Chr.	Interval	QTL	LOD	A	D	R2 (%)	
1	RM12210	qGL1	3.82	0.026	−0.017	3.96	
		qLWR1	5.09	0.011	−0.013	3.30	
2	Tw35293–RM266	qTGW2	11.76	0.39	0.15	14.50	
		qGL2	15.49	0.060	0.006	18.60	
		qGW2	6.15	0.013	0.006	5.69	
3	RM14302–RM14383	qGL3	3.81	0.025	0.027	4.01	
		qGW3	7.26	−0.014	−0.004	6.91	
		qLWR3	14.99	0.022	0.005	11.24	
4	RM16252–RM335	qTGW4	3.01	−0.18	0.06	3.41	
		qGW4	11.81	−0.017	−0.013	11.21	
		qLWR4	15.26	0.021	0.020	11.67	
5	RM3321–RM274	qGL5	17.79	0.062	0.005	21.67	
		qLWR5	11.31	0.018	−0.009	8.39	
6	RM549	qTGW6	6.58	0.27	−0.22	7.41	
		qGW6	5.82	0.012	−0.004	5.19	
7	RM10–RM70	qTGW7	12.15	0.39	−0.09	15.27	
8	RM22755–RM23001	qGW8	3.48	−0.009	−0.010	3.20	
		qLWR8	9.84	0.017	0.005	7.08	
9	RM219–RM1896	qGW9.1	3.70	0.009	0.012	3.88	
	RM107	qGW9.2	7.13	0.012	0.010	6.39	
10	RM1108–RM7300	qGW10	4.45	−0.004	−0.029	4.34	
		qLWR10	8.00	0.012	0.023	5.28	
11	RM167–RM287	qLWR11	7.58	−0.016	−0.019	7.05	
12	Tv963–RM27610	qGL12	3.85	0.031	0.015	4.95	
		qGW12	7.00	−0.013	−0.010	6.24	
		qLWR12.1	23.37	0.027	0.013	18.05	
12	Pita–RM511	qLWR12.2	4.59	−0.010	0.015	3.22	
Notes.

TGW 1,000-grain weight (g)

GL Grain length (mm)

GW Grain width (mm)

LWR Length/width ratio

A additive effect of replacing a Teqing allele with a IRBB52 allele

D dominance effect

R2 proportion of the phenotypic variance explained by the QTL

Five regions had significant effects on two traits. In the RM12210 region on chromosome 1 the IRBB52 allele increased GL and LWR by 0.026 mm and 0.011, respectively. In the RM3321−RM274 interval on chromosome 5, the IRBB52 allele increased GL and LWR by 0.062 mm and 0.018, respectively. In the RM549 region on chromosome 6, the IRBB52 allele increased TGW and GW by 0.27 g and 0.012 mm, respectively. In the interval RM22755−RM23001 on chromosome 8, the IRBB52 allele reduced GW by 0.009 mm and increased LWR by 0.017. In the interval RM1108−RM7300 on chromosome 10, the IRBB52 allele reduced GW by 0.004 mm and increased LWR by 0.012.

The remaining five QTLs were unaccompanied with other QTLs, including qTGW7 for TGW, qGW9.1 and qGW9.2 for GW, and qLWR11 and qLWR12.2 for LWR.

Dissection of two QTLs for grain size on chromosome 2

As described above, the terminal region of chromosome 2 had relatively large effects in terms of the number of QTLs detected and R2 of single QTL. Therefore, this region was chosen for further validation and fine-mapping.

Three NIL-F2 populations, ZH1, ZH2 and ZH3, were constructed, following the results of the W1 population (Fig. 3A). To fill the long distance between Tw32437 and Tw35293, two polymorphic markers, RM14034 and RM14056, were selected (Table S1). The two markers were homozygous in all the three populations. Two segregating regions, Tw31911−Tw32437 and Tw35293−RM266, were separated in the Z1 population (Fig. 3B). As shown in Table 2, QTLs were detected in both regions. In the interval Tw31911−Tw32437, the IRBB52 allele decreased TGW and GL but increased GW and LWR, having R2 of 6.05, 29.51, 13.61 and 28.52%, respectively. QTLs in this region affected GL and GW with opposite directions. The effect was larger on GL than on GW, resulting in the detection of a residual effect on TGW. Thus, this QTL was nominated as qGL2. In the Tw35293−RM266 region, the IRBB52 allele increased TGW, GL and GW, having R2 of 31.76, 17.95 and 3.73%, respectively. QTLs in this region affected GL and GW with the same direction, and the accumulative effect resulted in larger influence on TGW. Thus, this QTL was nominated as qTGW2.

Figure 3 Segregating regions of the eight populations.

(A) Three populations in F11. (B) Segregating regions of the three populations in F11 were updated with new polymorphic markers. Two sets of NIL populations in F11:12 were used to validate the genetic effects of qTGW2. (C) Three populations in F12 were used for fine-mapping of qTGW2.

Table 2 Dissection of two QTLs using three NIL-F2 populations.

Population	Segregating region	Trait	LOD	A	D	R2 (%)	
ZH1	Tw31911–Tw32437	TGW	2.83	−0.20	−0.15	6.05	
		GL	12.06	−0.069	−0.049	29.51	
		GW	5.77	0.004	−0.024	13.61	
		LWR	5.30	0.004	0.050	28.52	
	Tw35293–RM266	TGW	14.79	0.46	−0.19	31.76	
		GL	15.54	0.056	−0.008	17.95	
		GW	1.87	0.006	−0.003	3.73	
		LWR	0.17				
ZH2	Tw35293–RM266	TGW	17.60	0.55	−0.02	28.63	
		GL	13.15	0.065	−0.022	22.34	
		GW	13.54	0.021	−0.001	22.91	
		LWR	0.39				
ZH3	Tw35293–RM266	TGW	12.52	0.46	0.03	21.83	
		GL	17.42	0.061	−0.006	28.86	
		GW	12.81	0.020	0.001	22.37	
		LWR	0.38				
Notes.

TGW 1,000-grain weight (g)

GL Grain length (mm)

GW Grain width (mm)

LWR Length/width ratio

A additive effect of replacing a Teqing allele with a IRBB52 allele

D dominance effect

R2 proportion of the phenotypic variance explained by the QTL

The ZH2 and ZH3 populations were only segregated in the Tw35293−RM266 region. In both populations, significant effects were detected on all the traits except LWR. The enhancing alleles were always derived from IRBB52, and the effects were similar between the two populations. The additive effects were 0.55 and 0.46 g on TGW, 0.065 and 0.061 mm on GL, and 0.021 and 0.020 mm on GW. The R2 were 28.63 and 21.83% for TGW, 22.34 and 28.86% for GL, and 22.91 and 22.37% for GW. These results are similar to those found in the ZH1 population, indicating that qTGW2 located in the interval Tw35293−RM266 affects TGW, GL and GW with the same allelic direction. Since qTGW2 showed stable effects across the three populations, this QTL was selected for further analysis.

Validation and fine-mapping of qTGW2

Two sets of NILs, HY2 and HY3, were used to validate the genetic effects of qTGW2. Frequency distributions of the four traits were plotted using the two genotypic groups as two series (Fig. 4). For TGW, GL and GW, the difference between the TQ and IRBB52 homozygous genotypes were observed in both the populations. The IRBB52 homozygous lines were distributed in the higher-value region, and the TQ homozygous lines were distributed in the lower-value region. On the other hand, no distinction was found for LWR. These results indicate that QTLs for TGW, GL and GW were segregated in the two populations with the enhancing alleles derived from IRBB52.

Figure 4 Distributions of 1,000-grain weight, grain length and width, length/width ratio in HY2 and HY3.

(A–D) HY2. (E–H) HY3. NILTQ and NILIRBB52 are near isogenic lines (NILs) having Teqing and IRBB52 homozygous genotypes in the segregating region, respectively.

Results of the two-way ANOVA on the four traits are presented in Table 3. Highly significant effects (P < 0.0001) were detected for TGW, GL and GW in both the HY2 and HY3 populations. The effects were similar between the two populations, with the IRBB52 allele always increasing the trait values. The additive effects were 0.47 and 0.45 g on TGW, 0.054 and 0.061 mm on GL, and 0.021 and 0.013 mm on GW. The R2 were 60.94 and 66.02% for TGW, 48.26 and 62.12% for GL, and 36.67 and 22.74% for GW. In addition, significant influence on LWR were only detected in the HY3 population (P = 0.0003). The IRBB52 allele increased LWR by 0.009 with the R2of 12.94%. It was found that the allelic direction of qTGW2 remained unchanged across the five populations, with the IRBB52 allele always increasing TGW, GL and GW. As compared to the ZH1, ZH2 and ZH3 populations, the additive effect of qTGW2 hardly changed but the R2 values increased greatly in HY2 and HY3.

Table 3 Validation of qTGW2 using two NIL populations.

Population	Trait	Phenotype (Mean ± SD)	P	A	R2(%)	
		NILTQ	NILIRBB52				
HY2	TGW	25.23 ± 0.29	26.17 ± 0.34	<0.0001	0.47	60.94	
	GL	8.081 ± 0.045	8.189 ± 0.049	<0.0001	0.054	48.26	
	GW	2.784 ± 0.021	2.826 ± 0.025	<0.0001	0.021	36.67	
	LWR	2.915 ± 0.024	2.911 ± 0.031	0.5690			
HY3	TGW	22.65 ± 0.24	23.56 ± 0.26	<0.0001	0.45	66.02	
	GL	7.919 ± 0.041	8.040 ± 0.036	<0.0001	0.061	62.12	
	GW	2.652 ± 0.017	2.678 ± 0.018	<0.0001	0.013	22.74	
	LWR	3.000 ± 0.022	3.017 ± 0.015	0.0003	0.009	12.94	
Notes.

TGW 1,000-grain weight (g)

GL Grain length (mm)

GW Grain width (mm)

LWR Length/width ratio

A additive effect of replacing a Teqing allele with a IRBB52 allele

D dominance effect

R2 proportion of the phenotypic variance explained by the QTL.

NILTQ and NILIRBB52 Near-isogenic lines with Teqing and IRBB52 homozygous genotypes in the segregating region, respectively

To further narrow down the region of qTGW2, three polymorphic markers, RM14189, Tw35277 and Tw35395, were added (Table  S1). Three plants were identified from the W1 population and selfed to produce three NIL-F2 populations named Z1, Z2 and Z3. As shown in Fig. 3C, the segregating regions in Z1, Z2 and Z3 were RM14189−Tw35293, Tw35293−RM266, and Tw35395−RM266, respectively. QTL analysis for TGW, GL, GW and LWR were conducted (Table 4). Significant effects were detected in Z2 but not in the other two populations. This result suggests that qTGW2 was segregated in Z2 but not in Z1 and Z3. Consequently, the qTGW2 was delimited within a 103-kb region flanked by Tw35293 and Tw35395. In Z2, the IRBB52 allele increased TGW by 0.46 g, GL by 0.053 mm and GW by 0.014 mm, with the R2 of 29.93, 16.26 and 13.70%, respectively.

Table 4 Fine mapping of qTGW2 using three NIL-F2 populations.

Population	Trait	Phenotype (Mean ± SD)	LOD	A	D	R2 (%)	
		Teqing	IRBB52	Heterozygote					
Z1	TGW	24.03 ± 0.53	23.99 ± 0.51	23.94 ± 0.66	0.20				
	GL	8.017 ± 0.073	8.004 ± 0.072	8.004 ± 0.065	0.26				
	GW	2.805 ± 0.032	2.809 ± 0.036	2.808 ± 0.034	0.06				
	LWR	2.858 ± 0.026	2.850 ± 0.026	2.851 ± 0.025	0.65				
Z2	TGW	24.96 ± 0.54	25.90 ± 0.51	25.33 ± 0.49	18.38	0.46	−0.10	29.93	
	GL	8.156 ± 0.094	8.266 ± 0.091	8.205 ± 0.087	9.30	0.053	−0.008	16.26	
	GW	2.976 ± 0.026	3.004 ± 0.029	2.986 ± 0.022	7.76	0.014	−0.004	13.70	
	LWR	2.750 ± 0.041	2.762 ± 0.037	2.757 ± 0.031	0.68				
Z3	TGW	26.26 ± 0.88	26.34 ± 0.73	26.20 ± 0.84	0.22				
	GL	8.272 ± 0.111	8.262 ± 0.078	8.271 ± 0.113	0.07				
	GW	3.046 ± 0.046	3.054 ± 0.027	3.043 ± 0.046	0.51				
	LWR	2.726 ± 0.070	2.714 ± 0.029	2.727 ± 0.067	0.39				
Notes.

TGW 1,000-grain weight (g)

GL Grain length (mm)

GW Grain width (mm)

LWR Length/width ratio

A additive effect of replacing a Teqing allele with a IRBB52 allele

D dominance effect

R2 proportion of the phenotypic variance explained by the QTL

Discussion

Although progress has been made in fine mapping and cloning of major genes for grain weight, experimental constraints have limited our knowledge of minor genes that could be responsible for a larger proportion of trait variation. In this study, 27 QTLs for grain weight and shape in rice were detected using one population derived from an RH that was homozygous at major QTLs detected previously, followed by delimitation of qTGW2 for grain weight, length and width into a 103-kb region on chromosome 2.

All the populations used in this study were constructed from a single F7 plant of an indica rice cross. Among them, three NIL-F2 populations in F11 were grown under short-day conditions in Lingshui, and others were grown under long-day conditions in Hangzhou. qTGW2 showed stable effects on TGW, GL and GW across these populations. The IRBB52 allele increased TGW, GL and GW by a range of 0.45 to 0.55 g, 0.053 to 0.065 mm, and 0.006 to 0.021 mm, respectively. These results support that minor QTLs could be steadily detected in a highly isogenic background despite of diverse environment conditions, and the use of RHs could be an efficient way to detect and fine map minor QTLs.

Among the 16 QTLs cloned for grain weight and shape with major effects, GS3, OsLG3, OsLG3b, GS5, GSE5, GW6a, GL7/GW7, GLW7 and GW8 were found with high frequency in the modern rice varieties (Yan et al., 2009; Wang, Chen & Yu, 2011; Takano-Kai et al., 2009; Mao et al., 2010; Yu et al., 2017; Yu et al., 2018; Li et al., 2011; Duan et al., 2017; Wang et al., 2015a; Wang et al., 2015b; Wang et al., 2015c; Si et al., 2016. Two of the other QTLs, GW2 and GS2/GL2/GLW2 on chromosome 2 were rarely found in modern rice varieties. qTGW2 identified in this study was located in the interval Tw35293−Tw35395, corresponding to the 35.3−35.4 Mb region on the terminal end of chromosome 2, which was 6.4 Mb away from GS2 locus in the Nipponbare genome (IRGSP, 2005). The interval RM6−RM240 covering GS2 was detected as a non-segregating region in the populations used in the present study (Fig. 2). These results suggest that qTGW2 identified in this study is likely a new QTL for grain weight. Cloning and functional characterization of qTGW2 would provide new information for understanding the genetic and molecular basis of grain weight in rice.

Based on the Rice Genome Annotation Project (http://rice.plantbiology.msu.edu), there are 16 annotated genes in the 103-kb region for qTGW2 (Table S2). Thirteen of these genes encode known proteins, among which LOC_Os02g57630 encodes ubiquitin carboxyl-terminal hydrolase, LOC_Os02g57640 encodes a protein with the KH domain, LOC_Os02g57650 encodes a no apical meristem protein, LOC_Os02g57660 encodes phosphatidylinositol-4-phosphate 5-kinase, LOC_Os02g57670 encodes ribosomal L9, LOC_Os02g57690 encodes a kelch repeat protein, LOC_Os02g57700 encodes protein kinase, LOC_Os02g57710 encodes signal peptide peptidase-like 2B, LOC_Os02g57750 encodes a protein binding protein, LOC_Os02g57760 encodes O-methyltransferase, LOC_Os02g57770 encodes glycosyl hydrolases family 16, LOC_Os02g57790 encodes a ZOS2-19-C2H2 zinc finger protein, and LOC_Os02g57720 encodes an aquaporin protein. LOC_Os02g57720 may correspond to RWC3 and OsPIP2a. RWC3 was involved in the regulation of rice drought avoidance (Lian et al., 2004), and the expression of OsPIP2a in rapidly growing internodes of rice is not primarily controlled by meristem activity or cell expansion (Malz & Sauter, 1999). Of the remaining three annotated genes, LOC_Os02g57740 and LOC_Os02g57780 encode uncharacterized expressed proteins, and LOC_Os02g57730 encodes hypothetical protein. Further analyses are needed to confirm the candidate gene for qTGW2.

In addition to the qGL2−qTGW2 cluster on the terminal end of chromosome 2, a few other regions were detected to have important effects on grain weight and shape in the W1 population (Table 1). One of them, the RM1108−RM7300 region on the long arm of chromosome 10, has been targeted for more studies. Three QTLs were dissected (Zhu et al., 2019a), one of which was delimitated within a 70.7-kb region containing seven annotated genes (Zhu et al., 2019b). Five other regions, RM14302−RM14383 on chromosome 3, RM3321−RM274 on chromosome 5, RM10−RM70 on chromosome 7, RM167−RM287 on chromosome 11 and Tv963−RM27610 on chromosome 12, were previously reported to influence heading date differences between TQ and IRBB52 (Sun et al., 2018). Work is underway to determine the roles of these QTLs on multiple traits in rice.

Conclusions

A minor-effect QTL for grain weight, length and width in rice, qTGW2 located in the terminal region on the long arm of chromosome 2, was delimited to a 103-kb region flanked by Tw35293 and Tw35395 using NILs and NIL-F2 populations. This QTL had a consistent effect across different environment, providing a potential candidate gene for map-based cloning.

Supplemental Information

Table S1 The markers used in this study

Click here for additional data file.

Table S2 Annotated genes in the 103-kb region for qTGW2.

Click here for additional data file.

Supplemental Information 1 Raw data for QTL detection for four traits using the W1 population

Click here for additional data file.

Supplemental Information 2 Raw data for dissection of two QTL using three NIL-F2 populations

Click here for additional data file.

Supplemental Information 3 Raw data for validation of qTGW2 using two NIL populations

Click here for additional data file.

Supplemental Information 4 Raw data for fine-mapping of qTGW2 using three NIL-F2 populations

Click here for additional data file.

Additional Information and Declarations

Competing Interests

Author Contributions

Data Availability

The authors declare there are no competing interests.

Hui Zhang and Yu-Jun Zhu performed the experiments, analyzed the data, prepared figures and/or tables, authored or reviewed drafts of the paper, and approved the final draft.

An-Dong Zhu performed the experiments, prepared figures and/or tables, and approved the final draft.

Ye-Yang Fan and Ting-Xu Huang performed the experiments, authored or reviewed drafts of the paper, and approved the final draft.

Jian-Fu Zhang and Hua-An Xie conceived and designed the experiments, authored or reviewed drafts of the paper, and approved the final draft.

Jie-Yun Zhuang conceived and designed the experiments, analyzed the data, prepared figures and/or tables, authored or reviewed drafts of the paper, and approved the final draft.

The following information was supplied regarding data availability:

Data are available in the Supplementary Files.

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
