# Peer review of "Fine-mapping of qTGW2, a quantitative trait locus for grain weight in rice (Oryza sativa L.)"

_PeerJ, doi:10.7717/peerj.8679_

## Round 0.1 · original submission · Minor Revisions

The authors identified a novel QTL for grain weight, which lay a foundation for cloning and characterization of qTGW2 and marker-assisted breeding of high-yielding rice.

Reviewer 1 ·

Basic reporting

1、In the part of results “One QTL primarily contributed to grain shape, 182 especially to LWR, locating in the RM240−RM14034 region, namely, qGS2. Another QTL 183 simultaneously influenced TGW, GL and GW except LWR, flanking from RM14056 to the 184 terminal end of chromosome 2, namely, qTGW2”. The author should sequence the DNA of GS2 between the NILs and confirm whether qGS2 is the allelic of GS2, and the results must be submitted as attachment. Discarding the influence of qGS2 by getting rid of qGS2 fragment is essential to further studding the function of qTGW2 because GS2 positively regulates grain shape.
2、Besides of Tw35277 and Tw35395 in Table S1, sequences of other markers used in the study should also be provided, especially for the novel designed, which enable researchers to assess each QTL exactly and quote the paper accordingly.
3、The physical distance of Tw35293 and Tw35395 was 15kb in the Fig.3, while it was descripted as 103kb in the result, which is correct?
4、In the fig.3, the fig.A indicated that the physical distance is 400kb between the markers of Tw35293 and Tw32437, but in the fig.B, I found that 250kb and 1.9Mb was marked among this region. In total, the fig.3 should be redrawn because it is hard to understanding.
5、The loci of qTGW2, qGL2 and qGW2 were all located on the terminal end of the chromosome 2, just as showed in the Fig.2, why is the title named “Fine-mapping of qTGW2, a quantitative trait locus for grain weight in rice (Oryza sativa L.)”?
6、 “A total of 27 QTL were detected”. Here, “QTL” should be “QTLs”.
7、The first appearance of QTL should be descripted as quantitative trait loci.

Experimental design

no comment

Validity of the findings

no comment

Additional comments

This manuscript reported a novel QTL qTGW2 that regulate the development of grain size, which is a key agronomic trait controlling rice yield and quality and paid wildly attention both in rice biology and in rice breeding. The fine-mapping of qTGW2 is beneficial to isolate qTGW2 further and then to clear the molecular mechanism of rice seed development. Therefore, this study is advised to be significant. The questions must be answered in detail and the figures should be improved strongly.

Annotated reviews are not available for download in order to protect the identity of reviewers who chose to remain anonymous.

·

Basic reporting

This manuscript described detection of total of 27 QTLs for grain shape traits and fine-mapping one QTL ‘qTGW2’ by using segregating populations and near-isogenic lines (NILs) derived from across between two indica rice cultivars Teqing and IRBB. Genetic analyses revealed that there were two linked QTLs of ‘qGS2’ and ’qTGW2’ on the long arm of chromosome 2. The authors narrowed the candidate chromosome region of the one QTL ‘qTGW2’ within 105-kbp. This study clearly indicated importance of small effect QTLs involved in genetic architecture for quantitative traits such as grain shapes and yield components in rice. However, the authors should revise several descriptions in this manuscript.

Experimental design

Experimental design is suitable for getting the author's conclusion in this manuscript. However, the authors should revise several points described below.

1) L.94-97. The authors should describe names of the two crossing parents Teqing and IRBB to develop original RILs. And, the authors should describe the number of plants of W1 populations and the remaining eight populations.

2) L.213-217. The authors carried out QTL analysis in the Z2 population to confirm the ‘qTGW2’ region. However, you can find several plants having recombination between Tw35293 and RM266 in the Z2 population. If you check recombination points of each plants, you could delimit and narrow the present candidate QTL region about half length.

Validity of the findings

3) Although heritabilities of grain shape traits are generally higher than other agronomic traits, you should evaluate other agronomic traits such as heading date and plants height among the ZH1, ZH2, ZH3, HY2, HY3, Z1, Z2 and Z3 populations. If the authors observed no differences for other agronomic traits in the segregation populations, you should describe it at the Materials and Methods or the Results sections in this manuscript.

4) L.164-165, Table 1 and Figure 2. As you mentioned, the region on the terminal end of chromosome 2 showed larger genetic effect than other QTLs. However, several remaining QTLs also showed large genetic effects, for examples, QTLs on chromosomes 3, 4, 5, 7 and 12. The authors should also describe some discussion of these QTLs in the text.

·

Basic reporting

The English writing of Hui Zhang et al manuscript is professional and clear. In this MS, the previous research of grain weight traits in rice was described thoroughly, and the relevant literatures were cited properly. The structure of the article meets the requirements of the journal’s requirements, and the relevant figures and tables corresponding to the results of the article were presented in the article. The results of MS can explain the scientific problems concerned by the author and other potential readers in some extent.

Experimental design

The research topics of the MS fall in the journal’s scope and aims. The experimental design was well described and a rigorous analysis of the results was conducted in MS. The QTL qtgw2, which affects grain weight, was fine mapped by the authors, and the results enriched the genetic analysis of grain weight traits.
However, there are several points in materials and methods section need to be improved.
1. In the section of “field experiment and trace measurement”, the author has repeatedly described the number of samples with term 'approximately', which is not appropriate. The exact number of samples should be given.
2. In the section “DNA marker analysis”, the type and quantity of molecular markers should be described in details. Information of the molecular markers used in study shall be provided as supplementary materials.

Validity of the findings

qTGW2, which was identified by the authors, was not previously reported, and it affected both grain length and grain width. Furthermore, the author narrowed the region of qTGW2 loci within a 103kb interval and further analyzed the annotation genes in the region. The author provides sufficient data to ensure the reliability of the results, and conducted a good discussion on results.

Additional comments

1. The description in lines 86-91 should be modified. These sentences should demonstrate the purpose and significance of this study more than list the results.
2. Line190-191, ‘The IRBB52 homozygous lines were clustered towards to the high-value region, and the TQ homozygous lines were in the low-value region’, this sentence does not show the intended means, especially for the term of ‘high value region’ and ‘low value region’.
3. Some minor writing and grammatical errors should be carefully corrected. Line 108 'namely' should be 'named'.

---

## Round 0.2 · accepted · Accept

Your manuscript was reviewed carefully by myself and referred to reviewers for comment. On the basis of this study, I have decided to accept the manuscript for publication.

Reviewer 1 ·

Basic reporting

The author has well answered all questions I mentioned before, and I think the manuscript is suitable for the Peer J. While, the revise is still required to improve the manuscript quality by correcting some minor writing errors, such as “P441: BMC Genet should be BMC genetics”.

Experimental design

no comment

Validity of the findings

no comment

·

Basic reporting

This manuscript clearly revised by the authors based on the previous my comments. Author’s responses are reasonable for me. I understand and agree with the author’s responses.

Experimental design

No comment

Validity of the findings

No comment

Additional comments

No comment

·

Basic reporting

The writing of the MS is professional.The literature references are correct, and the results reach the self-containted level for the relevant results.

Experimental design

The research topic of the MS fall in the scope of the journal.
The methods used in MS are performed via a high technical standard and the results obtained are meaningful for understanding of rice grain.

Validity of the findings

The conclusion of the paper has been fully stated, which can answer the scientific questions concerned by the authors and potential readers.

Additional comments

All my concerns have been revised.